# Heart Histopathology and Mitochondrial Ultrastructure in Aged Rats Fed for 24 Months on Different Unsaturated Fats (Virgin Olive Oil, Sunflower Oil or Fish Oil) and Affected by Different Longevity

**DOI:** 10.3390/nu11102390

**Published:** 2019-10-07

**Authors:** María D. Navarro-Hortal, César L. Ramírez-Tortosa, Alfonso Varela-López, José M. Romero-Márquez, Julio J. Ochoa, MCarmen Ramírez-Tortosa, Tamara Y. Forbes-Hernández, Sergio Granados-Principal, Maurizio Battino, José L. Quiles

**Affiliations:** 1Department of Physiology, Institute of Nutrition and Food Technology “José Mataix Verdú”, Biomedical Research Center, University of Granada, Avda del Conocimiento sn., 18100 Armilla, Granada, Spain; mdnavarro@ugr.es (M.D.N.-H.); josemanuelromeromarquez@gmail.com (J.M.R.-M.);; 2UGC de Anatomía Patológica, Hospital San Cecilio de Granada, Avda, Conocimiento s/n, 18100 Granada, Spain; cesarl.ramirez.sspa@juntadeandalucia.es; 3Department of Biochemistry and Molecular Biology II, Institute of Nutrition and Food Technology “José Mataix Verdú”, Biomedical Research Center, University of Granada, Avda del Conocimiento sn., 18100 Armilla, Granada, Spain; mramirez@ugr.es; 4Nutrition and Food Science Group, Department of Analytical and Food Chemistry, CITACA, CACTI, University of Vigo, 36310 Vigo, Spain; tamaraforbe@gmail.com (T.Y.F.-H.); maurizio.battino@uvigo.es (M.B.); 5UGC de Oncología Médica, Hospital Universitario de Jaén, Avenida del Ejército Español 10, 23007 Jaén, Spain; sgranados@fibao.es; 6Genyo, Centre for Genomics and Oncological Research, Pfizer/University of Granada/Andalusian Regional Government, PTS Granada-Avenida de la Ilustración 114, 18016 Granada, Spain; 7Dipartimento di Scienze Cliniche Specialistiche ed Odontostomatologiche—Sez. Biochimica, Università Politecnica delle Marche, Ancona, 60131 Ancona, Italy; 8International Research Center for Food Nutrition and Safety, Jiangsu University, Zhenjiang 212013, China

**Keywords:** aging, histology, inflammation, fibrosis, mitochondrial morphology, MUFA, PUFA, n-3, n-6

## Abstract

Diet plays a decisive role in heart physiology, with lipids having especial importance in pathology prevention and development. This study aimed to investigate how dietary lipids varying in lipid profile (virgin olive oil, sunflower oil or fish oil) affected the heart of rats during aging. Heart histopathology, mitochondrial morphometry, and oxidative status were assessed. Typical histopathological features associated with aging, such as valvular lesions, endomyocardical hyperplasia, or papillary muscle calcification, were found at a low extent in all the experimental groups. The most relevant finding was that inflammation registered by fish oil group was lower compared to the other treatments. At the ultrastructural level, heart mitochondrial area, perimeter, and aspect ratio were higher in fish oil-fed rats than in those fed on sunflower oil. Concerning oxidative stress markers, there were differences only in coenzyme Q levels and catalase activity, lower in sunflower oil-fed animals compared with those fed on fish oil. In summary, dietary intake for a long period on dietary fats with different fatty acids profile led to differences in some aspects associated with the aging process at the heart. Fish oil seems to be the fat most protective of heart during aging.

## 1. Introduction

Cardiovascular diseases remain the first cause of death in the world and two of the main risk factors for its development are aging and diet [1,2]. Aging is a multifactorial process defined by an endogenous and progressive decay of the efficacy of the physiological processes over time [3,4]. This process is coupled with an increase in the susceptibility to suffer pathologies such as cancer, neurodegenerative or cardiovascular diseases [1]. One of the theories that try to explain aging is the free radical theory, which proposes biological aging caused by mitochondrial reactive oxygen species (ROS) production and subsequent damage [5]. Thus, mitochondria and oxidative stress have a relevant role in age-related pathological alterations, especially in the heart because it is the most mitochondrial-rich organ. Mitochondria provide ATP (adenosine triphosphate) through oxidative phosphorylation to be used by cardiomyocytes as the source of energy for maintaining the homeostasis and contractile function [4,6]. For this reason, the heart is particularly vulnerable to mitochondrial dysfunction. A general decline in mitochondrial function, clonal expansion of dysfunctional mitochondria, increased production of ROS (reactive oxygen species), suppressed mitophagy, and dysregulation of mitochondrial quality control processes are often implicated in cardiac aging and diseases [4]. These alterations can be at least in part due to changes in mitochondrial morphology by aging as well [7,8]. Intrinsic cardiac aging is determined by slow and gradual changes in heart physiological and morphological characteristics. Among these, action potential prolongation, changes in ionic exchanges across sarcolemma and dysregulation in calcium homeostasis are included. Also, increased fibrosis, mitochondrial defects and increased oxidative stress in cardiomyocytes are features of this process [9,10]. These changes make the heart more prone to stress, leading to increased alterations in the function and morphology of the organ which lately is responsible for the rise in cardiovascular mortality and morbidity with age [9,11]. 

Diet plays a determinant role in health maintenance and disease prevention [12]. Lipids are one of the three dietary macronutrients and, among these, the contributions of fatty acids stand out. Fatty acids can be classified in saturated or unsaturated relying on the absence or presence of double bonds in their hydrocarbon chain. In turn, unsaturated fatty acids can be divided into n-3, n-6 or n-9 based on the location of the first double bond counting from the methyl end. The most important monounsaturated fatty acids (MUFA) are n-9 and the main source is olive oil. Polyunsaturated fatty acids (PUFA) have been traditionally divided into n-3 and n-6. Concerning n-3, main sources are linseed, canola and fish oils. Meanwhile, n-6 PUFA comes from vegetable oils like sunflower or soybean. Dietary fatty acids upon ingestion are incorporated into biological membranes, including the mitochondrial membrane, contributing to cell structure and at the same time modulating many of its properties. In addition, they are utilized in energy production or converted into longer and more unsaturated fatty acids, which may give rise to biological compounds that may affect a variety of biological processes [13,14,15,16]. Besides, changes in mitochondrial fatty acid profile can modulate its susceptibility to oxidation in aging [17]. On other hand, and because the different effects on metabolism and membrane aspects, specific dietary fatty acids lead to different cardiovascular disease risks [18]. MUFA of the series n-9 present in virgin olive oil has shown a protective effect in heart mitochondria [15] and n-3 PUFA from fish oil have demonstrated a protective effect too, reducing cardiac morbidity and mortality [19]. However, n-6 PUFA from sunflower oil has manifested in aged heart mitochondrial damage [15], lower lifespan [20] and pro-inflammatory and pro-thrombotic effects [16,21]. Moreover, it was found that animals fed for life on virgin olive oil, sunflower oil or fish oil had different lifespan, with those fed on sunflower oil reaching a lower mean life and lower lifespan than those fed on virgin olive oil or fish oil [20].

The aim of the present study was to investigate different features at the heart of aged rats fed for 24 months on virgin olive oil, sunflower oil or fish oil as single dietary fats. In addition, we aim to increase our knowledge about how the fats investigated contribute to the survival of animals according to the effect they exert on each organ, in this case, the heart, during aging. This interest comes from the fact that it has been proved that the experimental fats tested in the present study, and with the same experimental conditions, give rise to variations in lifespan of rats, as stated above. In the present study, histopathology, mitochondrial morphometry, and different markers related to oxidative status have been analyzed at the heart. 

## 2. Materials and Methods

### 2.1. Animals and Diets

The animals were treated in accordance with the guidelines of the Spanish Society for Laboratory Animals and the experiment was approved by the Ethical Committee of the University of Granada (permit number 20-CEA-2004). Thirty-six male Wistar rats (*Rattus norvegicus*) weighing 80–90 g were housed and maintained in a 12 h light/12 h darkness cycle with free access to water. Diet was delivered *ad libitum* for the first two months and then, at 25 g/rat/day for the rest of the experiment (in order to avoid overweight). Rats were randomly assigned into three experimental groups and fed from weaning until 24 months of age on a semi-synthetic and isoenergetic diet according to the AIN93 criteria [22] modified in relation to the fat source (virgin olive oil, sunflower oil or fish oil). The fatty acid profile of experimental diets (% of the total fat) was as follows. For virgin olive oil, C14:0 (0.0), C16:0 (8.3), C16:1n9 (1.1), C18:0 (3.2), C18:1n9 (77.7), C18:2n6 (3.2), C20:3n6 (0.1), C20:4n6 (0.0), C20:5n3 (0.2), C24:0 (0.0), C24:1n9 (0.0), C22:6n3 (0.0). For sunflower oil, C14:0 (0.1), C16:0 (6.4), C16:1n9 (0.1), C18:0 (4.7), C18:1n9 (24.2), C18:2n6 (62.8), C20:3n6 (0.9), C20:4n6 (0.0), C20:5n3 (0.1), C24:0 (0.1), C24:1n9 (0.0), C22:6n3 (0.0). For fish oil, C14:0 (7.2), C16:0 (17.1), C16:1n9 (9.6), C18:0 (2.7), C18:1n9 (15.1), C18:2n6 (2.8), C20:3n6 (0.1), C20:4n6 (2.1), C20:5n3 (18.6), C24:0 (0.3), C24:1n9 (0.9), C22:6n3 (10.5). The night before sacrifice, animals fasted. Twelve rats per group were sacrificed by cervical dislocation followed by decapitation at 24 months from the beginning of the experiment. Six of them were used to pathological anatomy examinations and the other six, for the determination of the rest of the parameters. After exsanguination, heart was removed and preserved. 

### 2.2. Histopathological Analysis of Heart

Immediately after killing the animals, the heart was processed in its entirety by transversal cuts from the apex to the root of the great vessels with inclusion and subsequent staining with hematoxylin and eosin, following standard protocols. The evaluation of the possible cardiac injuries has been carried out by a specialist in pathological anatomy who worked blindly. The lesions have been assessed as shown below. In the case of the study of chronic progressive cardiomyopathy, inflammation and fibrosis have been evaluated. The inflammation was evaluated with a semi-quantitative scale ranging from 0 to V, being 0 when there were no lesions, I when there were less than three foci, II when there were more than three foci, III when the general inflammation was mild, IV when it was moderate and V when it was severe. In the case of fibrosis and lipofuscin depots, it was also assessed with a scale from 0 to V, being 0 when there were no lesions, I when there were less than three points of fibrosis or lipofuscin depots, II when there were more than three spots dotted or foci respectively, III when it was mild general fibrosis or depots, IV when it was moderate and V when it was severe. For vacuolization, a scale from 0 to IV was used, being 0 when no lipid degeneration was observed, I when there were minim depots, II when there were mild depots, III when there were moderate depots and IV when depots were severe. The calcification of the myocardium was assessed as absence or presence, as well as hyalinosis, the processes of mucinous degeneration and valvular hyperplasia. 

### 2.3. Electron Microscopy Analysis of Heart

A part of the fresh heart was processed for electron microscopy analysis. Briefly, sections were stained with uranyl acetate, counterstained with lead citrate, and viewed using a Carl Zeiss EM10C electron microscope (Oberkochen, Germany) at 4000×, 7500× and 40,000× magnifications. ImageJ 1.46r [23] was used for image analysis to determine the heart mitochondrial average area, perimeter, aspect ratio, circularity, and solidity. Area (μm^2^) is the surface selected in square units, perimeter (μm) is the length of the outside boundary of the selection, aspect ratio (AU) express the length-width relation (major axis/minor axis), circularity (AU) is a two-dimensional index of sphericity calculated as 4π·area/(perimeter)^2^, and solidity (AU) is defined by (area/convex area).

### 2.4. Heart Mitochondria Isolation

Heart (1 g) was used for mitochondria extraction following Fleischer et al. [24]. Heart mitochondrial protein was determined according to Lowry et al. [25]. 

### 2.5. Fatty Acids Analysis of Dietary Fats and of Heart Mitochondrial Membranes

The fatty acid profile of dietary fats and heart mitochondrial membranes was determined using the method of Lepage and Roy [26]. A gas-liquid chromatograph Model HP-5890 Series II (Hewlett Packard, Palo Alto, CA, USA) equipped with a flame-ionization detector was used to analyze fatty acids. Chromatography was performed using a 60-m-long capillary column, 32 mm id, and 20 mm thick impregnated with Sp^TM^ 2330 FS (Supelco Inc. Bellefonte, Palo Alto, CA, USA). The injector and the detector were maintained at 250 and 275 °C, respectively, nitrogen was used as a carrier gas, and the split ratio was 29:1. Temperature programming (for a total time of 40 min) was as follows: initial temperature, 160 °C for 5 min, 6 °C/min to 195 °C, 4 °C/min to 220 °C, 2 °C/min to 230 °C, hold 12 min, 1 °C/min at 160 °C.

### 2.6. Mitochondrial Hydrogen Peroxide Content 

The oxidation of the nonfluorescent 2’,7’-dichlorodihydrofluorescein diacetate (H_2_DCFDA), also known as dichlorofluorescin diacetate (DCFDA), to the highly fluorescent 2’,7’-dichlorofluorescein (DCF) was used to detect H_2_O_2_ content in heart mitochondria [27]. 

### 2.7. Heart Mitochondrial Antioxidants Enzymes Analysis

Catalase activity was determined following the method described by Aebi [28], by monitoring H_2_O_2_ decomposition at 240 nm, as a consequence of the catalytic activity of catalase. For cytosolic Se glutathione peroxidase (Se-GPx), we used the technique of Flohé & Günzler [29], a method based on the instantaneous formation of oxidized glutathione during the reaction catalyzed by glutathione peroxidase, which is continually reduced by an excess of active glutathione reductase and NADPH present in the cuvette. The subsequent oxidation of NADPH to NADP^+^ was monitored spectrophotometrically at 340 nm. Tert-butyl hydroperoxide was used as a substrate. Enzyme antioxidants were analyzed by adapting original methods for microplate reader performance. No more than 12 samples per bath were analyzed. Appropriate controls were used in order to avoid changes caused by differences in analysis conditions from batch to batch analysis.

### 2.8. α-Tocopherol and Coenzyme Q (CoQ) Determinations in Mitochondrial Membranes

CoQ_9_, CoQ_10_, and α-tocopherol in heart mitochondrial membranes were determined by HPLC attached to electrochemical detection, as previously described [30].

### 2.9. Statistical Analysis 

Results were expressed as mean ± standard error of the mean (S.E.M.) for six animals. Normal distribution and variance homogeneity were evaluated by Kolmogorov-Smirnov and Levene tests, respectively. Variables showing normal distribution were analyzed for differences between dietary treatments by an analysis of a ONE-WAY variance with a Bonferroni post hoc test. Non-normal variables were analyzed by Kruskal-Wallis and Mann-Whitney U non-parametric tests. Tamhane’s T2 test was applied to variables with non-homogeneous variances. In all analyses, significant differences were established at *p* < 0.05. Statistical analysis was performed with SPSS 24.0 for Windows (IBM, Chicago, IL, USA).

## 3. Results

### 3.1. Heart Mitochondrial Membrane Fatty Acids

Heart mitochondrial membrane fatty acid profile is presented in Table 1. No differences were found for the different fatty acids at the same treatment between both groups of age. Concerning differences between dietary treatments for each age, results were similar at 6 and 24 months of age. Summarizing, C18:1n9 (oleic acid, the most representative fatty acid found in olive oil) was located at the heart mitochondrial membrane in a higher percentage in animals fed with olive oil than other fed on sunflower oil or fish oil. For C18:2n6 (linoleic acid, the most representative fatty acid found in sunflower oil), the highest value was found in sunflower group followed by virgin olive oil and the lowest percentage was represented by the fish oil group. Finally, experimental treatment based on fish oil was the one that provided the highest percentage of C20:5n3 and of C22:6n3 (eicosapentaenoic (EPA) and docosahexaenoic acid (DHA), respectively, mainly in fish) in the heart mitochondrial membrane. In addition to the main fatty acids provided by each dietary treatment, significant differences were also observed in others. C18 amount was higher in fish oil group than in sunflower one. Concerning C20:4n6, the highest levels were found in animals fed on sunflower oil followed by those fed on virgin olive oil, the fish oil group showed the lowest values. Fish oil treatment, compared to the other two groups, led to the lowest levels of C24:1n9. 

### 3.2. Body and Heart Weight

At six months of age no differences were found between dietary treatments for the studied parameters. After twenty-four months of dietary treatment, bodyweight of animals fed on fish oil was significantly higher than those whose lipid intake was sunflower oil and similar to rats belonging to virgin olive oil group, as shown Table 2. Regarding the heart weight and heart/body ratio, there were no differences between treatments. No differences were found for food intake of the animals, which was indirectly monitored by food spillage and weekly measurement of body weight. The effect of aging was observed in the three investigated parameters, with young animals showing lower body and heart weights, but a higher heart to body weight ratio.

### 3.3. Histopathological Analysis

Histopathological analysis was systematically performed in the six animals of each group. No signs of pathology were found in any of the animals at six months of age. This is way only figures and numbers from 24-months old animals are showed in this section. Papillary muscle calcification was observed in three rats, two of them belonging to fish oil group and one of them to virgin olive oil group (Figure 1A). At valvular level, acellular material deposits rich in mucopolysaccharides were seen in almost every animal, but not accompanied by an inflammatory infiltrate (Figure 1B). The valvular lesions were scarce, existing in a punctual way in some rats. One of them, within sunflower oil group, suffered from endomyocardial hyperplasia (Figure 1C). Associated with chronic progressive cardiomyopathy, vacuolar myocardial degeneration was also observed on a regular time basis (Figure 1D). 

Undoubtedly, the most relevant result was the existence of chronic progressive cardiomyopathy, with the presence of inflammation and fibrosis in many animals. It was pointed out that the heart inflammation in old animals was significantly lower in those fed on fish oil than the other two groups while there were no differences between virgin olive and sunflower oil (Figure 2A,B). No statistically significant differences for heart fibrosis were found within fats, although the average trend showed higher levels in the sunflower oil group (Figure 2C,D). 

The incidence of different cardiac lesions in a semiquantitative scale ranging for each dietary treatment is illustrated in Table 3. The highlight was related to inflammation, where the fish oil group was the only one that contained an animal with no inflammation and where a lower grade was reached (grade II). On the other hand, the sunflower oil group registered the highest grade of inflammation (grade IV). Regard to fibrosis, the sunflower oil group was the only one in which all the animals presented the lesion, and that in which grade IV was reached. Concerning lipofuscin depots, a lower grade was observed in the fish oil group compared to the other two treatments. Coronary hyalinosis was slightly lower in the virgin olive oil group and vacuolization presented a higher grade in sunflower oil group than the other dietary fats.

### 3.4. Heart Mitochondrial Morphometry

Mitochondrial morphometry parameters are shown in Table 4. Representative pictures of different area, perimeter, aspect ratio, circularity, and solidity from heart mitochondria are shown in Figure 3. Concerning 6 months of age, no differences were found between dietary treatments. For 24 months old rats, the highest values for area, perimeter, and aspect ratio were found in the fish oil treatment, significantly different in comparison with the sunflower oil group. Regarding circularity and solidity, there were no differences between the three dietary treatments. In relation to age-related differences, area, perimeter and aspect ratio were lower for all dietary treatments at 6 months of age. Meanwhile, circularity and solidity were higher in 6 months old rats compared to those of 24 months. 

### 3.5. Oxidative Stress and Antioxidants Markers

Oxidative stress and antioxidant markers are collected in Table 5. For six-months-old rats, no differences were found between dietary treatments. Concerning 24 months of age, heart mitochondrial α-tocopherol and DCFDA values were similar in the three treatments. Heart cytosolic Se-GPX levels were higher in the virgin olive oil diet than in the sunflower or fish oil groups. The results pointed that the lowest catalase activity belonged to sunflower oil group, while there were no differences between virgin olive and fish oil groups. For the CoQ_9_, as well as in the case of CoQ_10_, significant differences were observed between the three treatments: the fish oil group exposed the highest value, followed by the virgin olive oil-fed animals and finally the sunflower oil group, having it the lowest value. Regarding differences between six and 24 months, all parameters were lower at six months of age, except for Se-GPX in fish oil fed animals and catalase in sunflower-fed rats were no differences were found.

## 4. Discussion

Life expectancy has increased in recent years and the world is moving towards an increasingly aged population [31]. Aging increases the risk of developing pathologies, among which those of heart-type. Cardiovascular diseases are related to oxidative stress status and lipid imbalance. So, modifications at this level could play an important role in the prevention and extent of these pathologies [32]. In this context, we investigated the effects of a treatment based on different dietary fats (virgin olive oil, sunflower oil, or fish oil) from weaning until the 24 months of age in rats. For that purpose, a complete histopathological analysis of the heart was realized accompanied by the analysis of mitochondrial morphometry as well as the evaluation of oxidative stress status. 

Prior to investigating the effects of experimental dietary fats intake on the aged heart of the rats, it is important to assess if fatty acids from experimental dietary fats have been incorporated into the biological membranes. Results from fatty acids lipid profile of heart mitochondria showed that animals fed on virgin olive oil registered the highest concentration of oleic acid. On other hand, rats fed on sunflower oil had the highest percentage of linoleic acid, and animals belonging to fish oil group showed the highest levels of eicosapentaenoic and docosahexaenoic acids. Something similar has been previously described concerning the administration of different unsaturated dietary fats to animals for long periods of time for the blood [33] and different tissues like liver [3,34], brain [35], skeletal muscle [36], and heart [15,37] of rats. Altogether these findings confirm that animals properly adapted to experimental dietary fats. Moreover, this fact demonstrated that the use of dietary fats with different fatty acids profile can be used as a therapy for the replacement of fatty acids in biological membranes during aging. Moreover, adaptation was effective from six months of age until 24 months of age. It deserved to be mentioned that experimental edible oils are a source of fatty acids (up to 99% of the composition are tryacilglycerides) but also possesses minor compounds like phenolics, tocopherols, and other antioxidants that in some extent may be responsible for the final effects of these oils in the heart of the animals. Nevertheless, since the main difference is related to fatty acid composition, discussion will be mainly focused on this aspect.

An interesting finding of the present study was that fish oil increased body weight in the rats. This is consistent with previous studies in mice [38] and rats [20]. Also, we have previously reported in the same animals from the present study a fat accumulation in different organs such as the pancreas, where fat infiltrations were observed [39] and in the liver, with higher fat depots after fish oil feeding [34]. Thus, the increase in body weight promoted by fish oil intake could be due to the fat accumulation in specific organs, although the hearts did not manifest different weights and the content of subcutaneous fat was not measured. Another cause could account for differences in fat mobilization speed from body depots depending on the fatty acid length and unsaturation number [40]. 

At the histopathological level and compared with young animals fed on the same treatments for six months (data not shown), all studied parameters were significantly higher in old animals. According to that, aging was associated with valvular lesions, endomyocardical hyperplasia, or papillary muscle calcification. According to the description and classification of valvular lesions in aged rats reported by Lewis [41], mucoid degeneration uses to be present in grade II and III, usually accompanied by inflammatory cells. Furthermore, its incidence and severity increase with age. In the present study, mucopolysaccharides deposits were found in almost every aged rat regardless the dietary treatment. However, an inflammatory infiltrate was not associated with that acellular material accumulation. Endomyocardical hyperplasia is characterized by the proliferation of fibroblast-like cells between endocardial endothelium and the cardiac muscle, predominantly found in the left ventricle. In the present research, this lesion was found only in one animal from the sunflower oil group. This represents around 5% of incidences, which is in agreement with previously described for male Wistar rats [41]. Papillary muscle calcification in the left ventricle was observed in three rats, two from fish oil group and one from virgin olive oil group. Small calcium deposits are a common finding in aged rats, mainly located in the apices of the papillary muscles and appear to have no functional consequences [42,43]. Lipofuscin depots, hyalinosis, and vacuolization were also analyzed, but there were no differences between the three dietary treatments. Lipofuscin amount increased with aging, but no correlation has been found between presence or amount of this pigment and clinically detectable heart disease [41].

Chronic progressive cardiomyopathy is the most frequent lesion in the heart of laboratory rats during aging. This lesion involves combinations of myofibrillar degeneration and/or necrosis with mononuclear cell infiltration and replacement fibrosis [41]. In the present study, heart fibrosis and inflammation were evaluated as markers of chronic progressive cardiomyopathy. Myocardial fibrosis is associated with cardiomyopathies development and increases with aging due to collagen accumulation [44,45,46]. Although our findings did not show statistical differences between dietary treatments, there was a trend of higher heart fibrosis in sunflower oil-fed animals. Indeed, it was the only group where grade IV of fibrosis (moderate) was found. On the other hand, the fish oil group inferred the highest number of animals with a total absence of fibrosis (grade 0), which could be related to the positive effect on interstitial fibrosis attributed to n-3 PUFA [47]. Weak differences in heart fibrosis found in the present study reported being statistically significant in the liver of the same animals [34]. As previously stated, myocardial inflammation uses to be induced by aging in the rat [44]. In the present study, dietary fat treatments led to differences in heart inflammation in the aged rats, with fish oil showing the lowest value. Anti-inflammatory and cardioprotective properties have been attributed to fish oil, especially because of its high content of n-3 fatty acids [48,49,50]. This effect could be due to the incorporation of fatty acids to cell membrane phospholipids that can be further transformed into prostanoids endowed with anti-inflammatory properties or less pro-inflammatory effect, or just because the promotion of a lower n-6/n-3 ratio in the membranes of animals fed on this dietary treatment [13,51]. Other mechanisms concerning the anti-inflammatory properties of n-3 PUFA have been described, such as the modulation of gene expression of mediators involved in inflammation [13] or reduced production of inflammatory cytokines [52]. 

Once the histologic aspects have been analyzed, it is interesting to investigate if changes at this level were associated with variations in the ultrastructure of heart mitochondria, which has been referred by other studies to be important when aging and dietary fat are considered [3,15,34]. Changes in mitochondrial ultrastructure are physiologically relevant, but also in relation to several pathologies [53]. Mitochondria are altered during aging due to an increase in oxidant production. Among typical changes associated with mitochondrial aging, they can be included the decline in membrane potential and the rise in peroxide production, among others [4,8]. Morphological changes at the mitochondria have been also observed with age, including higher length and more branched skeletal muscle mitochondria [54], larger mitochondria in vascular smooth muscle [55] and in heart [56]. Mitochondrial morphology is determined by fusion and fission processes and these events play important roles in maintaining the organelle homeostasis to ensure the correct function [57]. Indeed, a dysfunction in that mitochondrial dynamic is involved in cardiovascular diseases [58]. In the present study, age affected all investigated parameters, with no differences between dietary fats. Finding are in agreement with those previously described in relation to age, with bigger mitochondria in old animals. Dietary fatty acids differentially affected the heart mitochondrial morphology only in aged animals. These differences were found in relation to area, perimeter and aspect ratio, with significantly lower values in animals fed on sunflower oil than those fed on fish oil. No statistically differences were found between virgin olive oil and sunflower oil groups at 24 months of age, although a trend of higher values in the first was observed. Similar results were found in the heart of 24 months old rats fed on a diet with fat at 8% *w*/*w* [15], but, in that case, differences between the sunflower oil and virgin olive oil-fed animals were statistically significant, with less area and perimeter in sunflower oil group. According to results from both studies, fat amount is important when fat type is considered. It has been previously stated that increase in mitochondrial size is a feature of aging [8]. However, results from our laboratory and others indicate that this might be tissue and experimental conditions-dependent. Thus, we have found in the same animals from the present study increased mitochondria in aged animals at the liver, with those fed on sunflower oil showing a greater area and perimeter than those fed on virgin olive oil [34]. Frenzel and Feimann [59] found in the heart of Wistar rats that the mean size of mitochondria was decreased with aging. These authors also found that density of mitochondrial cristae was also markedly decreased. So, from the point of view of the heart and under our experimental conditions, aging leads to higher area and perimeter but depending on dietary fat this age-related increase may be different. 

As stated above, a feature of aging is oxidative stress [8]. In the present study, all investigated parameters were affected by age, with increased levels of damage in terms of H_2_O_2_ and higher values for most of the analyzed antioxidants. However, at 24 months, H_2_O_2_ levels did not show differences between treatments. On other hand, catalase activity was higher in virgin olive oil and fish oil groups compared to sunflower oil treatment, and Se-GPX was higher in virgin olive oil group. These findings are in agreement with other studies supporting, firstly, that age increases oxidative stress, and secondly that during aging, extra virgin olive oil treatment increases the activity of antioxidant enzymes in the liver [60]. Also, catalase activity in aged hearts of Wistar rats supplemented with fish oil has been reported to be higher compared with its control group [61]. Catalase and Se-GPX, involved in the removal of peroxides, it is well known that their activities increase with age [62]. Thus, treatment that infers a high activity of antioxidant enzymes would generate lower oxidative damage. Regarding lipophilic antioxidants in aged rats, α-tocopherol levels were similar between groups, while CoQ_9_ and CoQ_10_ were higher in rats fed on fish oil and the lowest values appeared in sunflower oil treatment. CoQ accumulation may reflect a higher antioxidant capacity of animals fed on fish oil. Altogether, higher antioxidant capacity resulting from the accumulation of CoQ plus increased catalase activity might be related to the lower inflammation found in the fish oil-fed rats compared with those fed on sunflower oil. Dietary treatments, based on the effects in the different organs or body compartment, can modulate the longevity and survival probability. Lifelong feeding on virgin olive oil or fish oil as in the present study reduced death probability compared to sunflower oil group in a parallel set of rats [20]. According to results from the present study, we can suggest that positive effects of fish oil on the heart of rats, and in a lower extent those of virgin olive oil feeding, might be responsible, at least in part, of the positive effects found on lifespan and longevity compared with animals fed on sunflower oil. 

## 5. Conclusions

The heart of aged rats registered a proper adaptation to the consumed diet, incorporating dietary fatty acids into the mitochondrial membranes. Features of heart in aging were observed in all groups, regardless of the treatment. However, fibrosis has shown a trend and inflammation was significantly lower in fish oil fed animals than in those fed on sunflower oil. Slight changes in ultrastructure and in oxidative stress status, with a higher antioxidant status in fish oil fed animals, might be related to the histopathological findings. These positive effects of fish oil on the heart of aged rats might be responsible for the well-known effects of n-3 PUFA on hearth physiology and the improved lifespan of rats fed on fish oil found in parallel studies.

## Figures and Tables

**Figure 1 nutrients-11-02390-f001:**
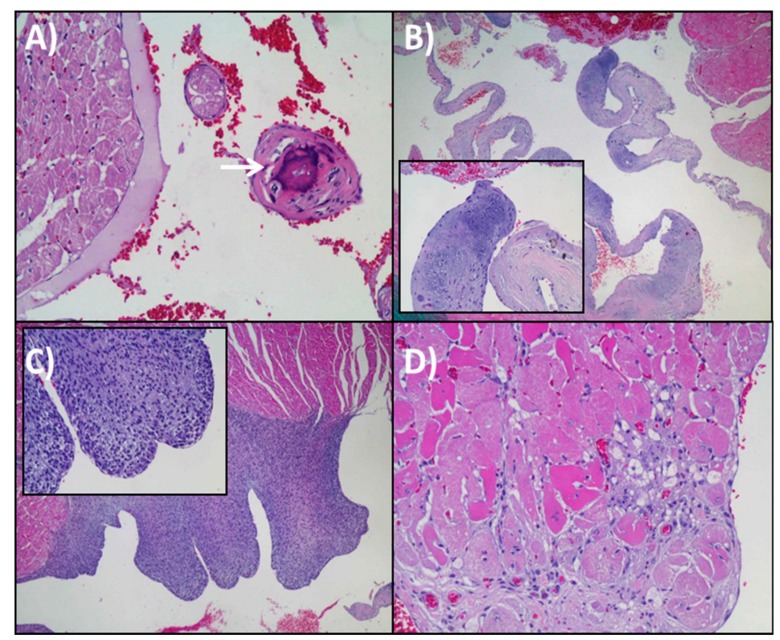
Representative images of histopathological features found in the heart of 24-months-old rats. (**A**) Papillary muscle calcification of left ventricle (white arrow). H & E 20×. (**B**) Mucinous degeneration in the cardiac valves. H & E, 4×. Insert 20×. (**C**) Endomiocardic hyperplasia. The proliferation of spindle-shaped cells. H & E, 4×. Insert, 20×. (**D**) Vacuolar degeneration (fat) at intramyocardial level. H & E 20×.

**Figure 2 nutrients-11-02390-f002:**
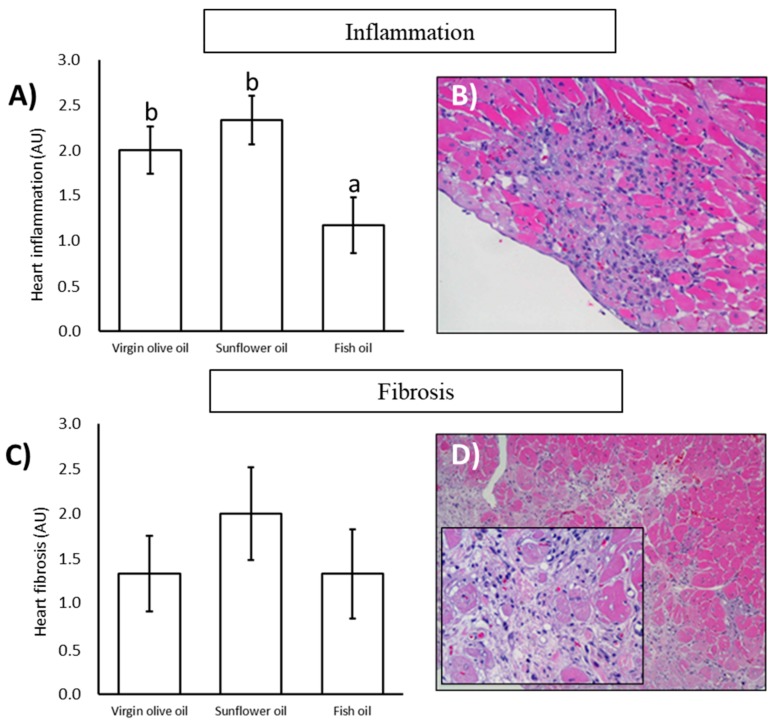
Inflammation and fibrosis in the heart of 24-month-old rats. (**A**) Data of semi-quantitative determination data of heart inflammation of rats fed on virgin olive oil, sunflower oil or fish oil. *n* = 6. (**B**) Representative image of a histological section of heart showing patches of inflammation. H & E, 20×. (**C**) Semi-quantitative analysis of heart fibrosis. *n* = 6. (**D**) Representative image of a histological section of heart showing fibrosis areas. H & E, 10×. Insert, 20×. For each parameter, lower-case letters, when different, represent statistically significant differences (*p* < 0.05) between the dietary treatments. Abbreviations: AU = Arbitrary Units.

**Figure 3 nutrients-11-02390-f003:**
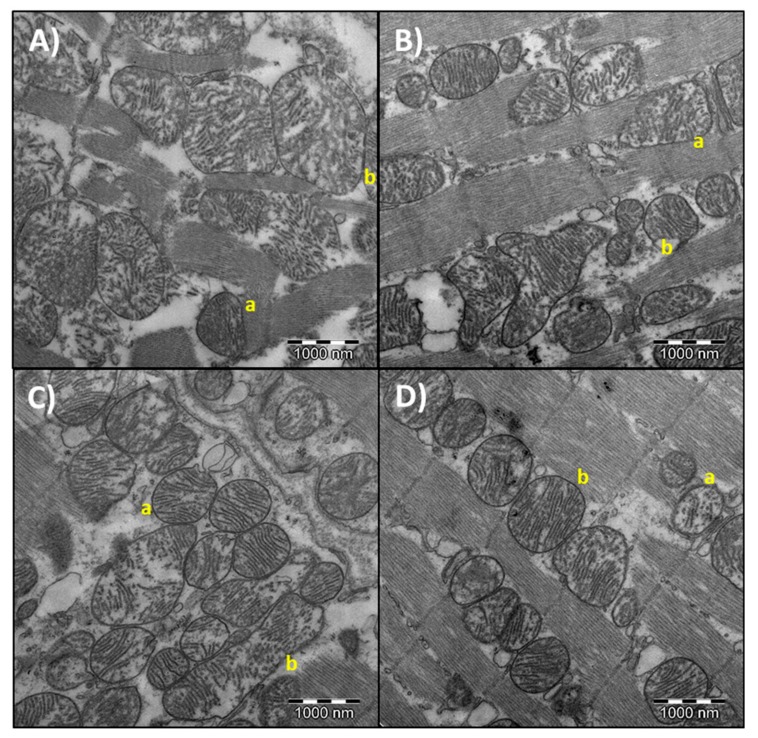
Representative images of mitochondrial ultrastructure in the heart of 6 and 24-month-old rats. (**A**) representative image of mitochondria with low (a) or high (b) area and perimeter. (**B**) representative image of mitochondria with low (a) or high (b) circularity. (**C**) representative image of mitochondria with low (a) or high (b) aspect ratio. (**D**) representative image of mitochondria with low (a) or high (b) solidity.

**Table 1 nutrients-11-02390-t001:** Heart mitochondrial membrane fatty acid profile (µg/mg) in rats fed for 6 or 24 months on virgin olive oil, sunflower oil or fish oil.

	Virgin Olive Oil		Sunflower Oil		Fish Oil	
	6 Months	24 Months	6 Months	24 Months	6 Months	24 Months
C14	201.7 ± 23.7	206.8 ± 72.2	191.3 ± 28.1	203.3 ± 30.0	151 ± 25.4	155.4 ± 49.8
C16	112.4 ± 25.1	126.0 ± 44.7	115.6 ± 21.9	126.1 ± 36.7	79.8 ± 18.4	75.4 ± 22.2
C16:1n9	3.61 ± 0.2	3.3 ± 0.4	2.5 ± 0.7	2.9 ± 0.8	3.2 ± 0.5	3.5 ± 0.8
C18	18.8 ± 3.25 ^A,B^	21.3 ± 4.6 ^a,b^	23.1 ± 1.1 ^B^	24.2 ± 1.2 ^b^	15.6 ± 2.5 ^A^	14.8 ± 2.9 ^a^
C18:1n9	21.9 ± 1.7 ^B^	21.2 ± 3.4 ^b^	16.8 ± 0.7 ^A^	15.7 ± 0.5 ^a^	10.1 ± 2.0 ^A^	12.2 ± 2.2 ^a^
C18:2n6	16.9 ± 1.8 ^B^	16.0 ± 2.0 ^b^	23.3 ± ±1.7 ^C^	24.4 ± 3.4 ^c^	5.5 ± 0.7 ^A^	5.9 ± 1.0 ^a^
C20:3n6	0.3 ± 0.1	0.4 ± 0.1	0.5 ± 0.2	0.5 ± 0.1	0.8 ± 0.2	0.8 ± 0.5
C20:4n6	11.7 ± 1.5 ^B^	10.7 ± 2.4 ^b^	15.4 ± 0.4 ^C^	16.0 ± 1.8 ^c^	7.1 ± 0.9 ^A^	6.1 ± 1.3 ^a^
C20:5n3	0.7 ± 0.2 ^A^	0.7 ± 0.3 ^a^	0.9 ± 0.5 ^A^	1.1 ± 0.7 ^a^	3.3 ± 0.6 ^B^	3.8 ± 1.5 ^b^
C22:6n3	6.8 ± 1.9 ^A^	7.2 ± 2.3 ^a^	7.1 ± 2.5 ^A^	8.1 ± 3.9 ^a^	15.9 ± 2.6 ^B^	17.3 ± 4.4 ^b^
C24:0	0.5 ± 0.4	0.3 ± 0.3	0.7 ± 0.3	0.9 ± 0.5	0.3 ± 0.2	0.4 ± 0.4
C24:1n9	1.9 ± 0.3 ^B^	1.3 ± 0.4 ^b^	2.7 ± 0.9 ^B^	3.2 ± 1.3 ^b^	0.3 ± 0.1 ^A^	0.3 ± 0.2 ^a^

Results are mean ± SEM for 6 animals. For each fatty acid, upper-case letter, when different, represent statistically significant differences (*p* < 0.05) between the dietary treatments at 6 months of age. For each fatty acid, lower-case letter, when different, represent statistically significant differences (*p* < 0.05) between the dietary treatments at 24 months of age.

**Table 2 nutrients-11-02390-t002:** Bodyweight (g), heart weight (g) and heart weight/body weight ratio of rats fed for 6 or 24 months on virgin olive oil or sunflower oil or fish oil.

	Virgin Olive Oil		Sunflower Oil		Fish Oil	
	6 Months	24 Months	6 Months	24 Months	6 Months	24 Months
Body weight (g)	309.2 ± 9.2 *	534.5 ± 51.8 ^a,b^	301.5 ± 10.9 *	508.3 ± 19.5 ^a^	315.3 ± 13.1 *	604.0 ± 23.3 ^b^
Heart weight (g)	0.9 ± 0.1 *	1.3 ± 0.1	0.9 ± 0.1 *	1.3 ± 0.1	0.9 ± 0.0 *	1.4 ± 0.1
Heart/Body ratio	0.0029 ± 0.0002 *	0.0024 ± 0.0001	0.0030 ± 0.0001 *	0.0026 ± 0.0000	0.0029 ± 0.0002 *	0.0023 ± 0.0001

Results are mean ± SEM for 6 animals. For each fatty acid, lower-case letters, when different, represent statistically significant differences (*p* < 0.05) between the dietary treatments at 24 months of age. * Differences for a particular parameter in the same dietary treatment between 6 and 24 months.

**Table 3 nutrients-11-02390-t003:** Incidence (%) of different cardiac lesions of rats fed for 24 months on virgin olive oil, sunflower oil or fish oil (*n* = 6).

	Virgin Olive Oil	Sunflower Oil	Fish Oil
	**Inflammation**		
Grade 0	0 (00.00%)	0 (00.00%)	1 (16.67%)
Grade I	1 (16.67%)	2 (33.33%)	3 (50.00%)
Grade II	4 (66.67%)	1 (16.67%)	2 (33.33%)
Grade III	1 (16.67%)	2 (33.33%)	0 (00.00%)
Grade IV	0 (00.00%)	1 (16.67%)	0 (00.00%)
	**Fibrosis**		
Grade 0	1 (16.67%)	0 (00.00%)	2 (33.33%)
Grade I	3 (50.00%)	3 (50.00%)	1 (16.67%)
Grade II	1 (16.67%)	1 (16.67%)	2 (33.33%)
Grade III	1 (16.67%)	1 (16.67%)	1 (16.67%)
Grade IV	0 (00.00%)	1 (16.67%)	0 (00.00%)
	**Lipofuscin depots**		
Grade 0	0 (00.00%)	1 (16.67%)	1 (16.67%)
Grade I	1 (16.67%)	1 (16.67%)	0 (00.00%)
Grade II	4 (66.67%)	3 (50.00%)	5 (83.33%)
Grade III	1 (16.67%)	1 (16.67%)	0 (00.00%)
	**Coronary hyalinosis**		
Absence	4 (66.67%)	3 (50.00%)	3 (50.00%)
Presence	2 (33.33%)	3 (50.00%)	3 (50.00%)
	**Vacuolization**		
Grade 0	1 (16.67%)	1 (16.67%)	1 (16.67%)
Grade I	5 (83.33%)	3 (50.00%)	5 (83.33%)
Grade II	0 (00.00%)	2 (33.33%)	0 (00.00%)

Semiquantitative scale. Inflammation: 0 = no lesions, I = less than three foci, II = more than three foci, III = mild, IV = moderate. Fibrosis: 0 = no lesions, I = less than three points of fibrosis, II = more than three spots dotted, III = mild, IV = moderate. Lipofuscin depots: 0 = no lesions, I = less than three foci, II = more than three foci, III = mild. Vacuolization: 0 = no lipid degeneration, I = minim depots, II = mild depots.

**Table 4 nutrients-11-02390-t004:** Heart mitochondrial morphometric parameters in rats fed for 24 months on virgin olive oil, sunflower oil or fish oil.

	Virgin Olive Oil		Sunflower Oil		Fish Oil	
	6 Months	24 Months	6 Months	24 Months	6 Months	24 Months
Area (µm^2^)	0.41 ± 0.01 *	0.72 ± 0.07 ^a,b^	0.38 ± 0.01 *	0.59 ± 0.04 ^a^	0.37 ± 0.01 *	0.97 ± 0.14 ^b^
Perimeter (µm)	2.65 ± 0.05 *	3.22 ± 0.17 ^a,b^	2.54 ± 0.05 *	2.93 ± 0.09 ^a^	2.58 ± 0.06 *	3.69 ± 0.22 ^b^
Circularity (AU)	0.90 ± 0.01 *	0.81 ± 0.01	0.88 ± 0.01 *	0.83 ± 0.01	0.90 ± 0.01 *	0.82 ± 0.03
Aspect ratio (AU)	1.23 ± 0.02 *	1.57 ± 0.07 ^a,b^	1.31 ± 0.02 *	1.56 ± 0.06 ^a^	1.28 ± 0.02 *	1.62 ± 0.19 ^b^
Solidity (AU)	0.97 ± 0.00 *	0.96 ± 0.00	0.97 ± 0.0 *	0.96 ± 0.00	0.97 ± 0.00 *	0.96 ± 0.00

Results are mean ± SEM for 6 animals. For each parameter, lower-case letter, when different, represent statistically significant differences (*p* < 0.05) between the dietary treatments at 24 months of age. * Differences for a particular parameter in the same dietary treatment between 6 and 24 months. Abbreviations: AU = Arbitrary Units.

**Table 5 nutrients-11-02390-t005:** Oxidative stress and antioxidants markers of rats fed for 24 months on virgin olive oil or sunflower oil or fish oil.

	Virgin Olive Oil		Sunflower Oil		Fish Oil	
	6 Months	24 Months	6 Months	24 Months	6 Months	24 Months
DCFDA (nM/mg)	2.79 ± 0.37 *	9.05 ± 1.52	3.59 ± 0.74 *	9.44 ± 1.27	1.66 ± 0.66 *	11.04 ± 1.78
α-tocopherol (µg/mg)	1.49 ± 0.18 *	3.99 ± 0.48	1.45 ± 0.10 *	3.94 ± 0.30	1.04 ± 0.12 *	4.15 ± 0.38
CoQ_9_ (µg/mg)	0.34 ± 0.08 *	3.18 ± 0.15 ^b^	0.29 ± 0.03 *	2.60 ± 0.18 ^a^	0.21 ± 002 *	4.18 ± 0.62 ^c^
CoQ_10_ (µg/mg)	0.05 ± 0.01 *	0.46 ± 0.06 ^b^	0.05 ± 0.02 *	0.33 ± 0.04 ^a^	0.04 ± 0.01 *	0.73 ± 0.09 ^c^
Se-GPX (U/mg)	435.94 ± 13.17 *	571.58 ± 9.62 ^b^	443.64 ± 15.66 *	520.27 ± 9.08 ^a^	500.89 ± 26.70	526.45 ± 19.30 ^a^
Catalase (U/mg)	36.61 ± 1.77 *	61.63 ± 1.40 ^b^	45.76 ± 5.89	53.13 ± 1.72 ^a^	47.82 ± 3.98 *	65.51 ± 3.14 ^b^

Results are mean ± SEM for six animals. For each parameter, lower-case letters, when different, represent statistically significant differences (*p* < 0.05) between the dietary treatments at 24 months of age. * Differences for a particular parameter in the same dietary treatment between six and 24 months. Abbreviations: DCFDA = dichlorofluorescin diacetate, CoQ = coenzyme Q, Se-GPX = Se glutathione peroxidase.

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
