# Peer review of "Heart Histopathology and Mitochondrial Ultrastructure in Aged Rats Fed for 24 Months on Different Unsaturated Fats (Virgin Olive Oil, Sunflower Oil or Fish Oil) and Affected by Different Longevity"

_nutrients, 2019, doi:10.3390/nu11102390_

Round 1

Reviewer 1 Report

Cardiovascular diseases are one of the major health concerns which are an off shoot of modern life style.The process of aging accelerates many deteriorating conditions associated with the functioning of the heart.Some of the parameters of concern are oxidative stress, declining antioxidant defenses, inflammation and mitochondrial dysfunction.It is known that some of these parameters can be manipulated by dietary lipids.In the present study the authors have chosen three dietary oils having different degrees of unsaturation and which has w9,w6 and w3 class of fatty acids.

Several changes were observed in the heart tissue as the rats grow older. But fish oil containing 18.6% EPA and 10.5% DHA were found to give some protection against these changes (biochemical and histopathological) in the heart of rats as they grow up to 24 months with fish oil feeding. Based on the results the authors concluded that positive effects of fish oil feeding on heart structure and function of aged rats may give an insight on the well established cardioprotective effects of w3 PUFA.

Author Response

Authors: The authors deeply acknowledge comments done by Reviewer.

Reviewer 2 Report

Summary:  The authors wanted to determine how long-term high fats diets of differing fatty acid composition affect cardiac histology and mitochondrial ultrastructure in the aging rat heart.  The fed rats diets containing of 1 of 3 polyunsaturated fats (PUFAs): virgin olive oil, sunflower oil, or fish oil for 24 months.  At the end of the feeding protocol rats were sacrificed and heart histopathology, mitochondrial morphometry, and oxidative status were assessed.  The concluded that different dietary fatty acid profiles lead to different age-related augmentations in the heart, with fish oil appearing to be the most protective. 

Comments:

1) Can the authors provide any insight to whether these diets altered mitochondrial metabolism?  (ie. other than what was reported regarding ox dative stress markers, was anything measured that may indicated that substrate oxidation was altered by any of the diets?)

2) The authors did not report the young cohort data, but by including it, it may strengthen the findings of this paper, particularly if the correct statistical anaylses are performed.

3) Were the rats fasted prior to sacrifice?

4) Presenting the fatty acids as a % of the total fat or even as a % of the total calories would be beneficial to the reader.

Author Response

Reviewer: Can the authors provide any insight to whether these diets altered mitochondrial metabolism?  (ie. other than what was reported regarding ox dative stress markers, was anything measured that may indicated that substrate oxidation was altered by any of the diets?)

Authors: The authors appreciate the suggestion of the Reviewer. The objective of this study was to evaluate the effects of the intake of different fat sources on pathological and ultrastructural aspects in the heart. Results related to oxidative status have also been included. Unfortunately, the authors have not performed any metabolic studies on this experimental design. However, Reviewer's recommendation will be taken into account for future studies.

Reviewer: The authors did not report the young cohort data, but by including it, it may strengthen the findings of this paper, particularly if the correct statistical analyses are performed.

Authors: Following Reviewer’s recommendation, authors have included data on 6 months of age for all the studied parameters. This was not possible only for data on histopathology since no pathological findings were found for rats at 6 months of age. In other words, young animals had no pathological findings so all values for 6 months-old rats were 0 in relation to these parameters.

Reviewer: Were the rats fasted prior to sacrifice?

Authors: The rats were fasted the night before being slaughtered. This information has now been included in the material and methods section (page 3 line 115).

Reviewer: Presenting the fatty acids as a % of the total fat or even as a % of the total calories would be beneficial to the reader.

Authors: Following Reviewer’s recommendation authors have changed the information concerning this issue (page 3, line 109).

Round 2

Reviewer 2 Report

No further comments.

This manuscript is a resubmission of an earlier submission. The following is a list of the peer review reports and author responses from that submission.

Round 1

Reviewer 1 Report

These authors studied the effects on aged rats’ heart supplemented with different unsaturated fatty acids. The results were not well organized and did not match with relative figures. This reviewer is fully confused the figure content. The results are not robust to support any conclusion.

Concerns:

 1, why the authors only detected 3 fatty acids in the heart in figure 1. The membrane fatty acids profile includes a list of fatty acids. The figure 1 with low quality. Please fill different pattern or color in different bar. Animal number should be shown.

2, I don’t understand the figure 2 at all. Which panel is oil fish group? Why the authors selected 3 rats? It's very hard to read the results associated with the figure 2. What about the results of sunflower oil group? Scale bar is missing.

3, Again, in figure 3.  Which group did the panel B show? And panel D. The B and D should  match with A and C.

4, Also in figure 4. Each panel representants for what?

5, the title must be modified with scientific writing.

6, Discussion is too long.  

Author Response

Reviewer: 1

Comments and Suggestions for Authors

These authors studied the effects on aged rats’ heart supplemented with different unsaturated fatty acids. The results were not well organized and did not match with relative figures. This reviewer is fully confused the figure content. The results are not robust to support any conclusion.

Concerns:

Reviewer: Why the authors only detected 3 fatty acids in the heart in figure 1. The membrane fatty acids profile includes a list of fatty acids. The figure 1 with low quality. Please fill different pattern or color in different bar. Animal number should be shown.

Authors: Mitochondrial fatty acids have been used to verify a proper adaptation of the animals to the diet. Although the fatty acids profile of heart mitochondrial membrane was fully characterized, authors initially decided to present in the manuscript one the most representative fatty acid of each type of fat (C18:1n9 for virgin olive oil, C18:2n6 for sunflower oil and C22:6n3 for fish oil). Even so, based on reviewer’s suggestion, Figure 1 has been replaced by Table 1, which collects the fatty acid profile.

Animal number has been specified, according reviewer’s recommendation in tables and figures and a sentence has been added to the Mat & Meth issue as follows:

-Page 3, lines 116-118: “Six of them were used to pathological anatomy examinations and the other six, for the determination of the rest of the parameters”.

Reviewer: I don’t understand the figure 2 at all. Which panel is oil fish group? Why the authors selected 3 rats? It's very hard to read the results associated with the figure 2. What about the results of sunflower oil group? Scale bar is missing.

Authors: Figure 2, now named Figure 1, does not collect a summary of different experimental groups (virgin olive oil, sunflower oil, or fish oil). As specified in the figure legend, the figure content is referring to representative findings in aged rats (papillary muscle calcification, mucinous degeneration in the cardiac valves, endomiocardic hyperplasia with proliferation of spindle-shaped cells and vacuolar degeneration (fat) at intramyocardial level) regardless the experimental group where they were observed. For authors, it is relevant to show an example of these finding, but, as described in the text; these findings did not appear systematically in all rats of all groups. For that reason, a figure cannot show the average aspect of a given findings in a comparative way in the different experimental groups. Notwithstanding, authors have tried to improve slightly the legend description in order to avoid misunderstandings.    

Concerning reviewer’s question about why author have selected three rats, this is not correct. Histopathological analysis was performed systematically in the six rats of each group. Authors believe that the reviewer refers to page 6 line 207 where is written: “Papillary muscle calcification was observed in three rats…” That sentence does not mean that calcification has been studied in three rats, but that it only appeared in three animals of eighteen (six per group) considered. In order to avoid misunderstandings, authors have modified the beginning of the histopathological results description to clarify that the study were performed in six rats per group, as follow:

-Page 6, line 209: “Histopathological analysis was systematically performed in the six animals of each group.”

Regarding reviewer’s commentary about scale bar, the scale has been indicated in the figure legend, expressed as the specific magnification (4x or 20x) following international standards of graphical representation of histopathological results.   

Reviewer: Again, in figure 3. Which group did the panel B show? And panel D. The B and D should match with A and C.

Authors: As the previous answer, Figure 3, now named Figure 2, does not collect a summary of different experimental groups (virgin olive oil, sunflower oil, or fish oil). This figure shows a representative image of a histological section of heart inflammation and fibrosis. It is not a representative image of findings in a particular dietary group, but a general manifestation of these parameters regardless the experimental group. For authors, it is relevant to show an example of these finding. According to reviewer’s comment, authors have tried to improve slightly the legend description in order to avoid misunderstandings.

Following reviewer’s recommendation, authors have modified Figure 3, now Figure 2, in order to math panel A and C with B and D, respectively.        

Reviewer: Also in figure 4. Each panel representants for what?

Authors: In the same line of previous answers, Figure 4, now named Figure 3, collects general representative images of heart mitochondrial ultrastructure: area and perimeter, circularity, aspect ratio and solidity. In other words, this figure is not a representation of mitochondrial morphometry parameters of each dietary treatment, but a general representation of mitochondrial morphometry parameters described. For authors, it is relevant to show an example of these features. According to reviewer’s comment, authors have tried to improve slightly the legend description in order to avoid misunderstandings.       

Reviewers: The title must be modified with scientific writing.

Authors: Following Reviewer’s recommendation, the title of the manuscript has been changed as follows:

-Page 1, lines 2 to 5: “Heart histopathology and mitochondrial ultrastructure in aged rats fed for 24 months on different unsaturated fats (virgin olive oil, sunflower oil or fish oil) and affected by different longevity”

Reviewer: Discussion is too long.  

Authors: Authors have tried to reduce the extension of the discussion based on reviewer’s comment.

Reviewer 2 Report

I believe this data would be better presented with the longevity data and data from other tissue rather than a stand alone paper.

Author Response

Reviewer: 2

Comments and Suggestions for Authors

I believe this data would be better presented with the longevity data and data from other tissue rather than a stand alone paper.

Authors: The authors appreciate the reviewer’s suggestion. However, longevity data and data from other tissues or organs of the animals used in the present study such as liver or pancreas have already been published (reference number 20, 34 and 39 of manuscript, respectively).

Reviewer 3 Report

Aging is an inevitable process for living beings. This is accompanied by many biochemical and physiological changes. The wellbeing and longevity often depends on how well the individual handles the metabolic changes that occur during aging process. Heart is a vital organ which comes under attack by extraneous factors which include oxidative stress, failing antioxidant defense mechanisms, inflammation and histopathological changes associated with aging. Mitochondrial integrity is another pointer for cardiovascular health. It is known that some of these parameters are influenced by dietary components particularly the lipids that can influence the extent of changes which may have bearing on cardiovascular health. The work reported in the present manuscript brings out the impact of consuming oils with different composition with emphasis on w9, w6 and w3 fatty acids on changes in heart integrity. The results indicated that rats fed a diet containing virgin olive oil, sunflower oil, fish oil for a period of 24 months incorporated the major fatty acids from these oils into mitochondrial membrane lipids. Fish oil feeding significantly lowered histopathological changes, mitochondrial morphometry and also enhanced antioxidant defense systems to lower oxidative stress which were the results of aging process. Based on the results obtained the authors concluded that fish oil provides positive protection to heart during aging.

Minor clarifications to be addressed:

1.      Rats fed fish oil showed higher gain in body weight (Table 1). Did rats receive diet ad libitum or was it pair fed? What was the feed intake in different groups of rats? Food efficiency ratio (gain in body weight/ feed consumption) will give a better picture about the impact of dietary lipids on body weight gain. Calorie restriction also has an impact on longevity.

2.      Line 111-Fish oil did contain EPA (18.6%) and DHA (10.5%) w3 fatty acids. Hence incorporation of EPA in mitochondrial membrane may also be indicated.

3.      Even though it is not the objective of this study, it is known that oils do contain minor constituents which have independent health benefits (eg. Olive oil contains polyphenols which function as nutraceuticals). Is there any possibility that such minor constituents may also have benefits on cardiac health?

Author Response

Reviewer: 3

Comments and Suggestions for Authors

Aging is an inevitable process for living beings. This is accompanied by many biochemical and physiological changes. The wellbeing and longevity often depends on how well the individual handles the metabolic changes that occur during aging process. Heart is a vital organ which comes under attack by extraneous factors which include oxidative stress, failing antioxidant defense mechanisms, inflammation and histopathological changes associated with aging. Mitochondrial integrity is another pointer for cardiovascular health. It is known that some of these parameters are influenced by dietary components particularly the lipids that can influence the extent of changes which may have bearing on cardiovascular health. The work reported in the present manuscript brings out the impact of consuming oils with different composition with emphasis on w9, w6 and w3 fatty acids on changes in heart integrity. The results indicated that rats fed a diet containing virgin olive oil, sunflower oil, fish oil for a period of 24 months incorporated the major fatty acids from these oils into mitochondrial membrane lipids. Fish oil feeding significantly lowered histopathological changes, mitochondrial morphometry and also enhanced antioxidant defense systems to lower oxidative stress which were the results of aging process. Based on the results obtained the authors concluded that fish oil provides positive protection to heart during aging.

Minor clarifications to be addressed:

Reviewer: Rats fed fish oil showed higher gain in body weight (Table 1). Did rats receive diet ad libitum or was it pair fed? What was the feed intake in different groups of rats? Food efficiency ratio (gain in body weight/ feed consumption) will give a better picture about the impact of dietary lipids on body weight gain. Calorie restriction also has an impact on longevity.

Authors: Rats received paired fed diets. Since the rats are social animals, they should be kept in groups and not in individual cages, which greatly hindered the monitoring of food intake individually. Notwithstanding, controlled amounts of diet were daily provided in any cage: 25 g/rat/day. No significant amounts of remained food or spillage were found in any cage. Moreover, a good way to monitor food intake is the control of body weight gain, which was done every week. This information allowed the authors to infer that food intake was not different between animals or groups. For this reason, food efficiency was considered the same in all groups. Authors have added this information as follow:

-Page 3, lines 104-106: “Diet was delivered ad libitum for the first two months and then, at 25 g/rat/day for the rest of the experiment (in order to avoid overweight).”

-Page 6, lines 206-207: “No differences were found for food intake of the animals, which was indirectly monitored by food spillage and weekly measurement of body weight.”

Reviewer: Line 111-Fish oil did contain EPA (18.6%) and DHA (10.5%) w3 fatty acids. Hence incorporation of EPA in mitochondrial membrane may also be indicated.

Authors: As reviewer suggests, incorporation of EPA in heart mitochondrial membrane is significantly higher in rats fed on fish oil than those treated with virgin olive oil or sunflower oil. In order to avoid a loss of information, authors have included now the complete fatty acid profile by changing Figure 1 by Table 1.

Reviewer: Even though it is not the objective of this study, it is known that oils do contain minor constituents which have independent health benefits (eg. Olive oil contains polyphenols which function as nutraceuticals). Is there any possibility that such minor constituents may also have benefits on cardiac health?

Authors: Authors totally agree with Reviewer in the appreciation about the putative effect of minor components. This may happen in some extent for the three edible oils, and it is the result to work with a whole food instead with a pure mixture of fatty acids. In fact, virgin olive oil has a minor fraction that is rich in phenolic compounds, among others; sunflower oil is added with up to 1200 ppm of a-tocopherol to avoid rancidity, and the same happens with fish oil in relation to a-tocopherol, retinol and others. So, in order to answer the Reviewer concern, authors have added a sentence in the manuscript as follows:

-Page 9, lines 269 to 273: “It deserved to be mentioned that experimental edible oils are a source of fatty acids (up to 99% of the composition are tryacilglycerides) but also possesses minor compounds like phenolics, tocopherols and other antioxidants that in some extent may be responsible for the final effects of these oils in the heart of the animals. Nevertheless, since the main difference is related to fatty acid composition, discussion will be mainly focused in this aspect.”